# Automatic speech recognition predicts contemporaneous earthquake fault displacement

Christopher W. Johnson [1,3] ✉, Kun Wang[1,2,3] & Paul A. Johnson[1]

Significant progress has been made in probing the state of an earthquake fault by applying machine learning to continuous seismic waveforms. The breakthroughs were originally obtained from laboratory shear experiments and numerical simulations of fault shear, then successfully extended to slow-slipping faults. Here we apply the Wav2Vec-2.0 self-supervised framework for automatic speech recognition to continuous seismic signals emanating from a sequence of moderate magnitude earthquakes during the 2018 caldera collapse at the Kīlauea volcano on the island of Hawai'i. We pre-train the Wav2Vec-2.0 model using caldera seismic waveforms and augment the model architecture to predict contemporaneous surface displacement during the caldera collapse sequence, a proxy for fault displacement. We find the model displacement predictions to be excellent. The model is adapted for near-future prediction information and found hints of prediction capability, but the results are not robust. The results demonstrate that earthquake faults emit seismic signatures in a similar manner to laboratory and numerical simulation faults, and artificial intelligence models developed for encoding audio of speech may have important applications in studying active fault zones.

Aggregate signals contained in acoustic emissions, identified when applying machine learning to laboratory stick-slip experiments, demonstrate the contemporaneous timing and state of the loading cycle is observable directly from the continuous data[1,2]. Numerous successes of these observations are documented when applying machine learning to map the continuous acoustic emission data from laboratory shear experiments to a target label of the fault displacement, friction, fault gouge thickness, etc.[3–10], with results that extend to physics-informed approaches[11,12] and the timing of near-future predictions of slip[8,13–15]. Applying the laboratory developed techniques to slow-slipping events in the Cascadia subduction zone, that recur about every 15 months, led to successful predicting the displacement rate and time-to-failure from continuous seismic waveforms[16,17]. The build up and release of stress for this subduction zone occurs on time scales captured by modern seismic and global navigation satellite system

(GNSS) instrumentation, and provides an excellent case study for transitioning from the laboratory scale to fault zones. The elusive step is translating the laboratory analogy to seismogenic stick-slip fault motion associated with earthquakes on active fault systems that produce strong, damaging ground motions[18].

The primary challenge for applying machine learning to frictional failure on most faults hosting large magnitude earthquakes is the long repeat times, ranging from decades to thousands of years, and therefore, the lack of geophysical instrumental data that captures a complete loading cycle instead of a small fraction of a loading cycle. Attempts to address this problem at the laboratory scale include training deep learning models with data from numerical simulations that represent the laboratory shearing apparatus and apply transfer learning to account for the dearth of data representing a complete loading cycle[19]. The transfer learning approach opens the possibility

[1]Los Alamos National Laboratory, EES-17 National Security Earth Science, Los Alamos, NM 87545, USA. [2]Now at ExxonMobil Technology and Engineering Company, Energy Sciences Research Division, Los Alamos, NJ, USA. [3]These authors contributed equally: Christopher W. Johnson, Kun Wang. ✉e-mail: cwj@lanl.gov

for assessing the timing of fault slip where data only exists for a small portion of the earthquake loading cycle[19]. The proper development of deep learning transfer models, applicable to earthquake faults, requires high-fidelity geophysical measurements to produce a foundational framework for extracting salient information from the continuous waveforms. This effort requires targeted case studies that investigate repeating earthquake failure on time scales short enough to contain the entire cycle recorded with modern instrumentation. One such case study and an advanced self-supervised machine learning model framework is presented here.

In 2018 the Kīlauea volcano on the island of Hawai'i began erupting and resulted in a sequence of more than 50 collapse events on the caldera-ring-fault with displacement equivalent to approximately $M_w 5$ earthquakes[20]. The earthquake sequence is well recorded with a permanent network of instrumentation (Fig. 1) that captures the inflation and deflation of the caldera during a 3 month period[20–23]. This sequence of repeating moderate magnitude earthquakes offers an excellent case study to determine if the advancements developed in the lab[15] are applicable to this tectonic environment.

In our previous work, supervised learning gradient boosted decision tree models applied to time-series features derived from continuous seismic data at the Kīlauea volcano are able to predict the GNSS surface displacement and time-to-failure of the next ring-fault-collapse event[24]. These contemporaneous predictions are performed

using spectral and temporal features calculated from 30 seconds of waveforms and are analogous the laboratory modeling results that capture the instantaneous state of the system[1,2]. However, due to nonstationarity in the sequence of collapse events[23], a standard moving time-window analysis applying gradient boosted trees was insufficient to produce robust results. To address the nonstationarity[24], blocked the data into multi-day windows to provide a complete distribution of signals in the training set. This treatment of the data, while also applying protocols to ensure no timing information was present in the data, produced a model that could adequately reproduce the observed ground displacements but highlighted the limitations of developing supervised learning data sets. The inability of gradient boosted tree models to properly generalize due to the nonstationarity data is problematic when trying to extract salient features for downstream tasks, as is done with a foundation model framework.

In the present work we address this and apply a self-supervised learning approach to determine if contemporaneous and future predictions can be made directly from the continuous seismic data. The widespread adoption of unsupervised, or self-supervised, learning techniques for signal classification has not seen the same rise in popularity for seismic waveform analysis as compared to other deep learning fields; despite intriguing results demonstrating the ability of ML models to separate signals with no prior knowledge[25–27]. The lack of generalization in supervised learning models does not facilitate fine-

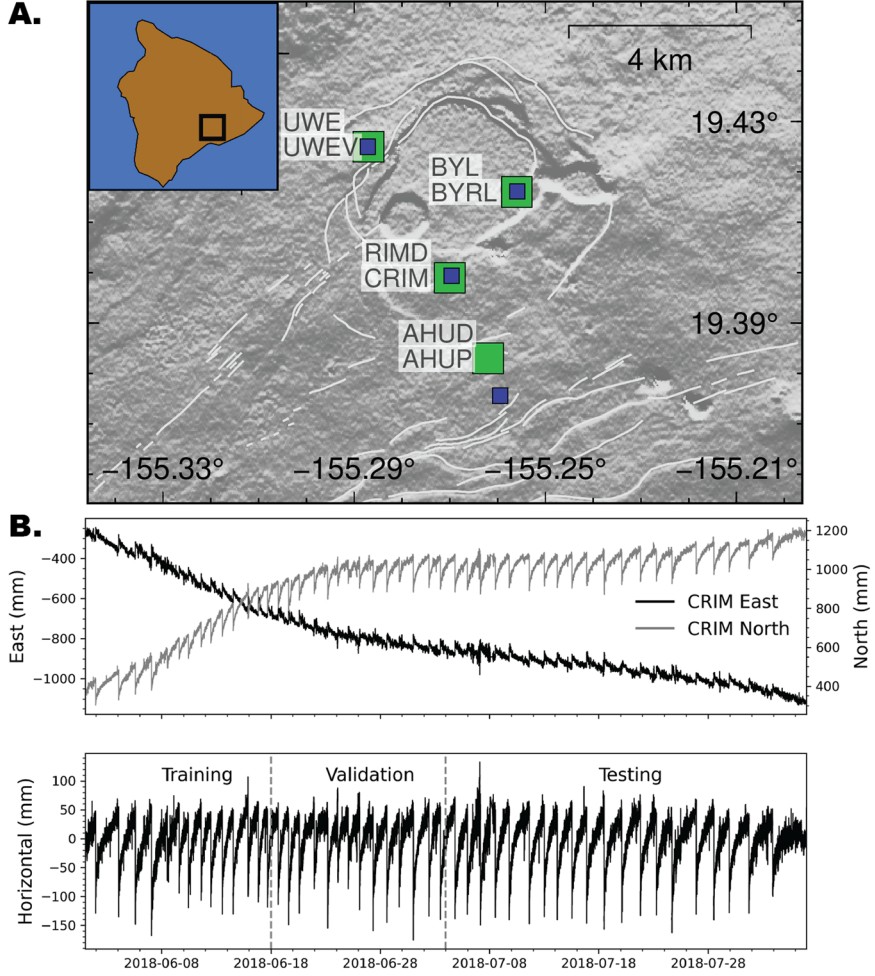

**Fig. 1 | Study area and data. A** Kīlauea volcano located on the island of Hawai'i (see inset) is shown centered on the caldera with white lines for the mapped faults, green squares indicate the GNSS station locations, and blue squares are the col-located seismic stations. **B** The ground motions during the inflation and deflation of the collapse events are analyzed in this study. Shown is the GNSS horizontal

displacement time series for station CRIM, located on the caldera rim, for the north and east displacement, and the combined horizontal with the long-period behavior removed that is used for the model. The time periods for the training, validation, and testing are listed.

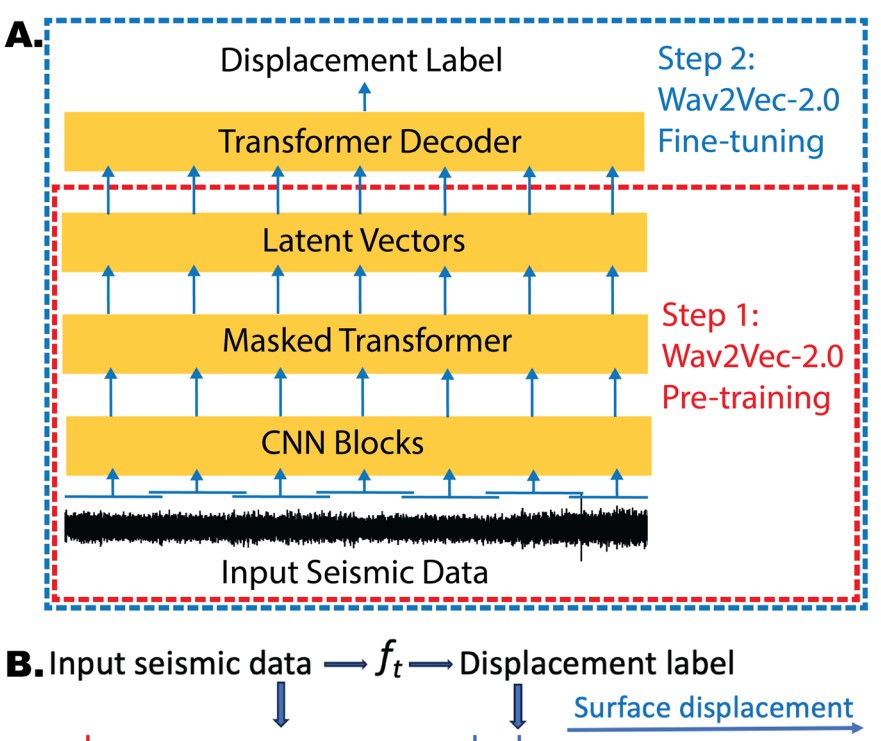

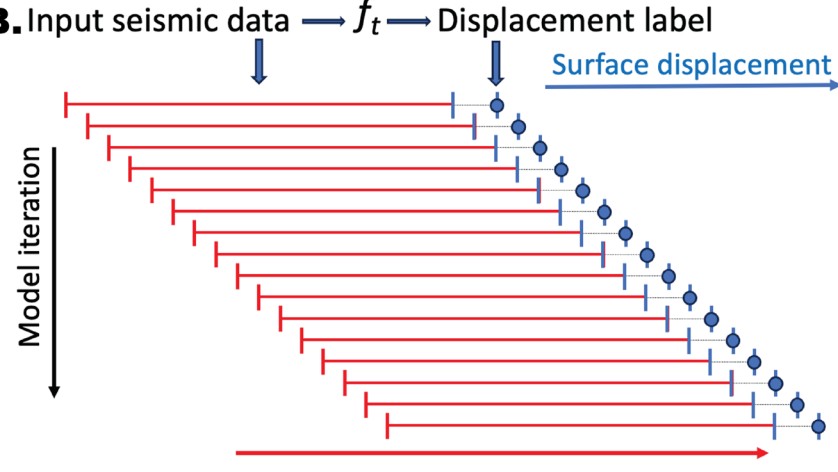

**Fig. 2 | Diagram of the modified Wav2Vec-2.0 model that maps the seismic waveform to fault displacement. A** The input passes through a stack of convolutional layers that extract hierarchical features and then input to a masked transformer encoder to capture long-range dependencies in the data, allowing the model to learn complex patterns and relationships within the seismic signals, then to a transformer decoder to predict the label. **B** Conceptual approach of the model input passing successive time-windows of seismic data to the Wav2Vec-2.0 model. With each iteration the model attempts to map characteristics of the seismic signal onto latent space vectors and then to successive windows of the contemporaneous GNSS displacement. The mapping is a regression to a single value equivalent to the horizontal displacement magnitude in the time window. With each successive iteration the model steps sequentially through later time windows and attempts to map to a later displacement value.

tuning an existing model for out-of-distribution data, and instead requires large amounts of labeled training data to develop a new model for a specific application[28]. Training models using supervised learning techniques is now eclipsed by the development of self-supervised foundation models using unlabeled data to produce generative pre-trained transformer networks[29]. Transformer networks, popularized by natural language processing applications, are comprised of attention blocks that enable developing connections in the data with long range dependencies[30] without the problems associated with long series of input data. Transformer models show great potential for geophysical time series data sets[6,15,19,31]. A convolutional-encoder-decoder transformer model, developed by us[15], learns a latent space codebook that contains information extracted from the input seismic waveforms and is specifically designed for contemporaneous fault displacement. The success of our previous approach motivates the augmentation of our model with a more advanced waveform

encoder architecture developed for automatic speech recognition applied to the Kīlauea volcano data for contemporaneous ground displacement predictions and the onset of a major slip event. The results presented provide insight to the required signal encoding that is necessary for foundation modeling efforts applied to seismogenic earthquake applications.

## Results
### Wav2Vec-2.0 deep learning for predictions
The modeling framework applied is the Wav2Vec-2.0 architecture[32] developed for self-supervised learning of latent representations from raw audio waveforms by Facebook AI Research (FAIR). It is designed to encode the input signals that convert spoken language into written text, a task referred to as Automatic Speech Recognition[33]. The Wav2Vec-2.0 model was originally developed to learn representations of audio data without the need for labeled training data by applying a

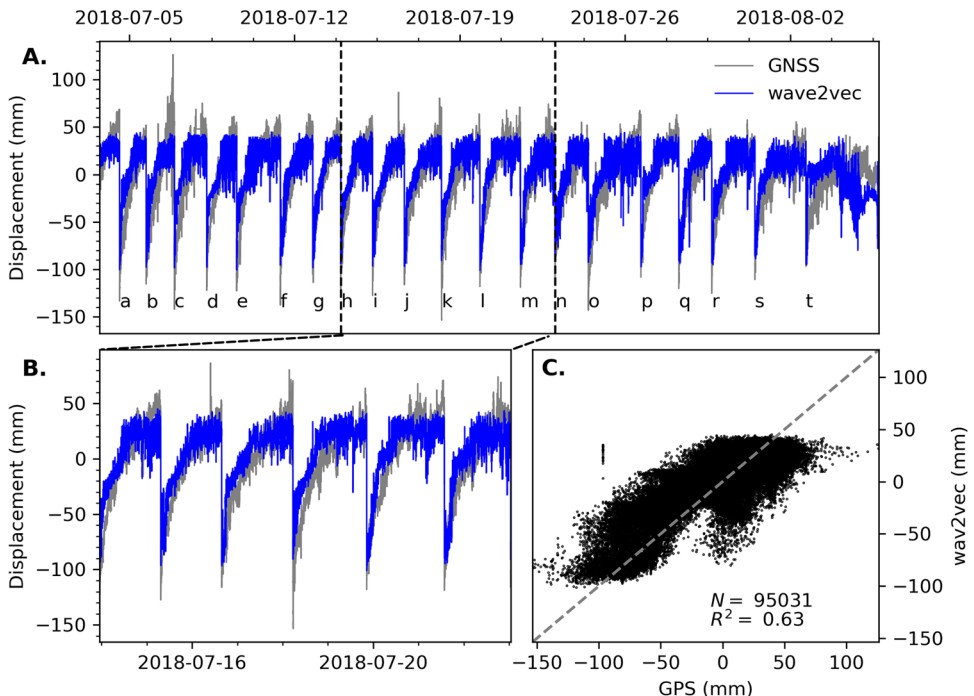

**Fig. 3 | Contemporaneous displacement prediction.** The modified Wav2Vec-2.0 model maps the continuous seismic signal to the fault displacement. **A** The model predictions are in blue and the GNSS data are in grey. For this time-window pair, the model employs 300 seconds of continuous seismic data as input and maps to a single GNSS displacement value that is temporally equivalent to a single value occurring in the final 30s of the seismic data window, i.e., 270-300 seconds of the seismic data stream. **B** A subset of nine days are shown in more detail to highlight the model output. **C** The $R^2$ is 0.63 for the testing with an upper limit in the predictions visible in the correlation plot.

self-supervised learning approach where the model is trained on a large dataset of unlabeled audio clips. The model employs a "context window strategy" to process long audio sequences by dividing the input audio into overlapping segments, each with a fixed length (Fig. 2). This allows the model to capture information from a broader context, which is important for understanding patterns in speech.

As applied here, the model inputs are continuous seismic waveforms, that are analogous to audio data, and the model is trained to predict contemporaneous fault displacement. In brief, the model design employs a stack of convolutional encoder layers that extract hierarchical features from the raw seismic waveforms. The hierarchical feature extraction captures information at different levels of abstraction, enabling the model to learn both low-level and high-level representations across a broad frequency spectrum of the input signal. After the convolutional encoder layers, the model uses transformer blocks that are known for their effectiveness in capturing long-range dependencies in sequential data. The combination of convolutional and transformer blocks allow the model to learn complex patterns and relationships within the datastream. For dimension reduction the model incorporates vector quantization to discretize the continuous embeddings produced by the model, making them more manageable while preserving important information. These embedding representations are used for various downstream tasks, e.g., speaker identification.

The self-supervised nature of the training process allows Wav2Vec-2.0 to learn from a large amount of unlabeled data. Each waveform example is encoded by the convolutional layers to a sequence of latent vectors. Then approximately half of these vectors are masked before going through the transformer encoder. The pre-training task is for the model to restore the missing vectors correctly from a quantized codebook of vectors. At each masked location, the model needs to distinguish between the true answer (positive example) and another 100 distractors (negative examples) uniformly sampled from other positions. Using a cosine similarity contrastive loss, the model learns to bring its restored vectors closer to the corresponding true target vectors in the feature space, while pushing the distractors farther apart. The combination of self-supervised learning, advanced feature extraction techniques, and contextualized learning contributes to the model success in accurately transcribing spoken language into written text. The modeling framework is particularly useful in scenarios where labeled audio data is scarce or expensive to obtain, analogous to continuous seismic and geodetic data for the typical timescales of earthquake faults.

This modeling application has not been applied to continuous seismic waveform data to our knowledge, but is well suited for the task addressed here. The input signal is comprised of seismic waveforms sequences and the target variable is the corresponding high-rate GNSS derived surface displacement. A conceptual diagram illustrating the model and how the mapping between the input seismic data and the surface and fault displacement is shown in Fig. 2. An in-depth description of the deep learning model design, the seismic and GNSS data sets, and the training procedures is provided in the Methods section.

## Wav2Vec-2.0 model predictions

**Contemporaneous displacement prediction.** A range of input seismic and output displacement time-window pairs are tested to identify the window pair that optimizes the displacement prediction. For example, 300 seconds of seismic data are used as a model input that maps to a single surface displacement during the final 30 second interval (from 270-300 seconds). Note the 5 second GNSS data is resampled to 30 seconds for model training and testing (see Methods). Following the prediction, the input and output windows are shifted by 30 seconds and the procedure is repeated (Fig. 2B). The model is tested for input and output time windows ranging from 30-300 seconds (Supplementary Fig. S1). The grid search results show

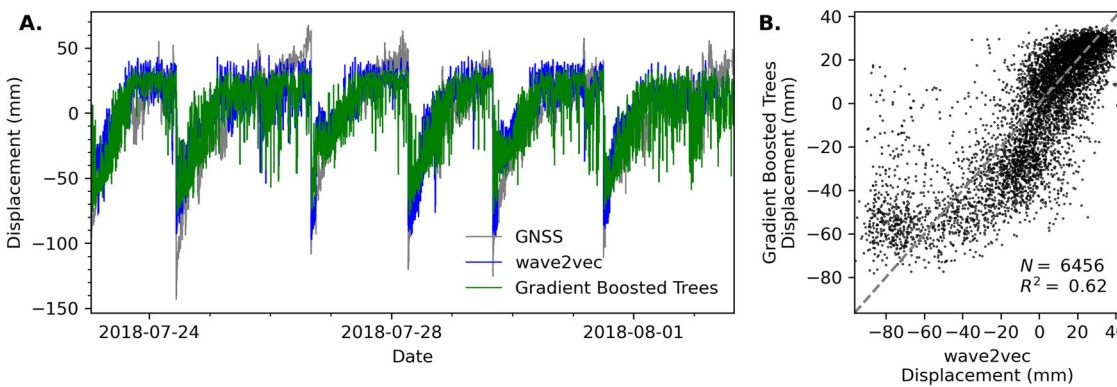

**Fig. 4 | Comparison of Wav2Vec-2.0 GNSS displacement predictions to a gradient boosted tree model**[24]**. A** In blue is Wav2Vec-2.0, in green is the gradient boosted tree, and in gray is the GNSS. **B** Correlation plot for the Wav2Vec-2.0 and gradient boosted tree results with a $R^2$ value of 0.62 for the 6456 common time points.

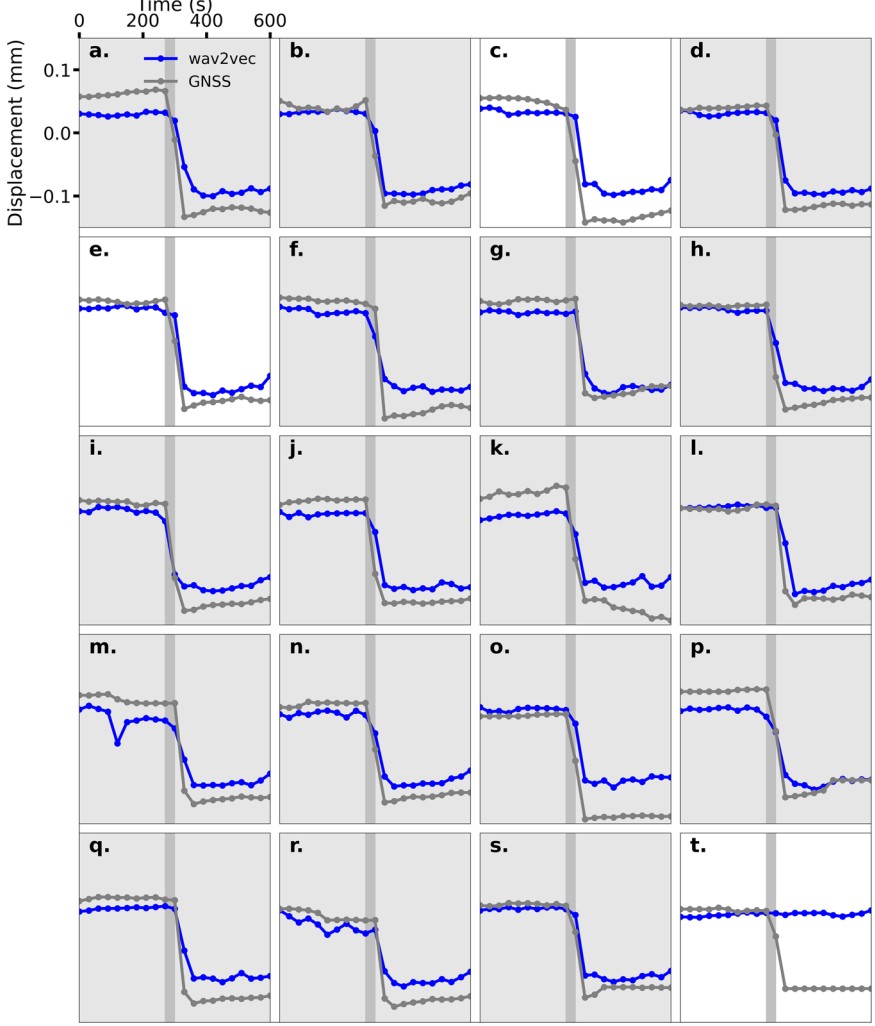

**Fig. 5 | Timing of the onset of slip for the collapse events.** Model predictions for the onset of slip for 20 collapse events where the panels (**a**–**t**) corresponds to the labels in Fig. 3. Each panel shows a 600 second window with a dot each 30 seconds for the time-steps. In blue are the model results and in gray are the GNSS measurements. The vertical dark gray bar is the expected window to see a drop in displacement during the onset of slip. The background is shaded for the 17 events that the model correctly predicts.

the input window size of 300 seconds with an output window size of 30 seconds produces good results, and input window sizes of 240 and 150 seconds are only slightly better. Indeed nearly all input window lengths with output of 30 seconds produced positive results (Supplementary Fig. S1).

The results using seismic waveform data from station RIMD to predict the GNSS station CRIM surface displacement nicely track the ground motions through the deformation cycle (Fig. 3A). Viewing 9 days of the displacement prediction shows the steady increase throughout the cycle, preceding the collapse event (Fig. 3B) with a

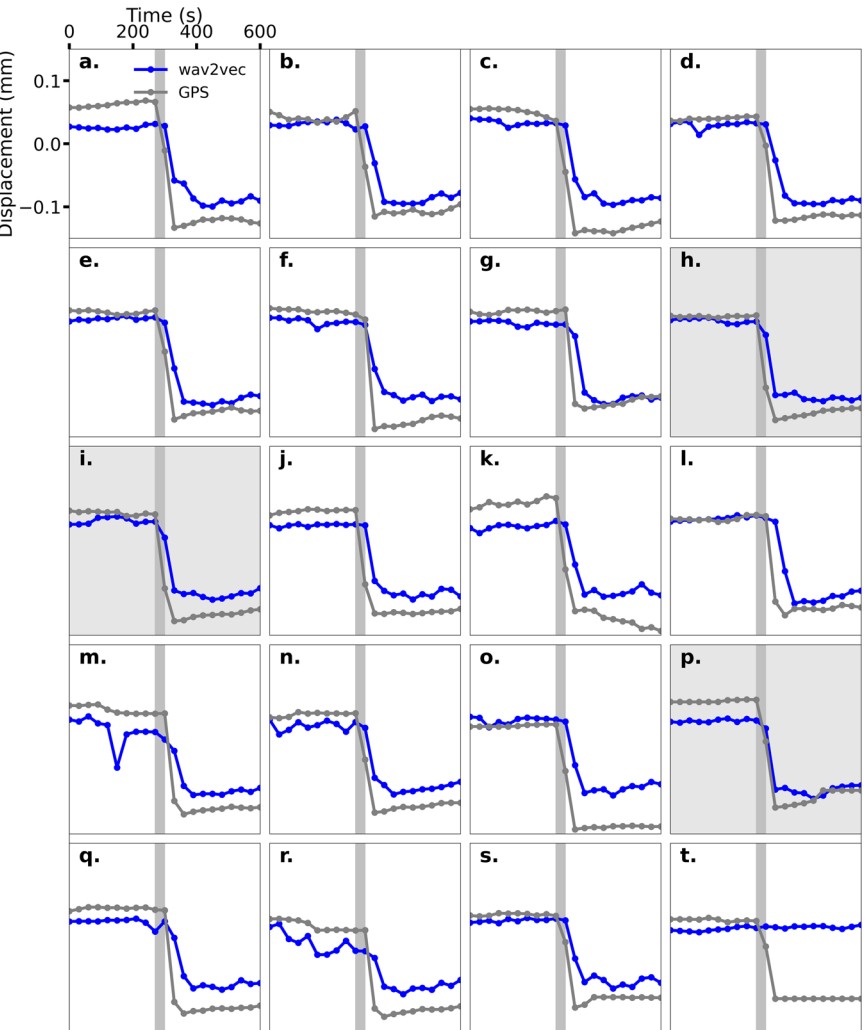

**Fig. 6 | Future prediction for the onset of slip for the collapse events.** Future model predictions for the onset of failure for 20 collapse events where the panels (**a**–**t**) corresponds to the labels in Fig. 3.The model inputs 300 seconds of history to then predict the GNSS displacement at the next time step. The vertical dark gray bar is the expected window to see a drop in displacement during the onset of slip. The background is shaded for the 3 events that the model correctly predicts the future failure event.

0.63 $R^2$ value for the entire series (Fig. 3C). Overall, the predictions are better than the gradient boosted tree approach previously applied[24]–in that work the $R^2$ was 0.59 for the same station pair (Fig. 4A).

Quantitatively the increase in $R^2$ could be considered quite modest, qualitatively we show a large improvement in the range of values at the lower displacement prediction which occur following a slip event (Fig. 4B). One would not expect the $R^2$ value to solely capture the performance change considering the overall variance in the training data (Fig. 1) and the inherent point-to-point fluctuations in GNSS high rate measurements. The assessment of improvement from the gradient boosted tree model involves a combination of factors. The comparison of the 2 different model techniques shows similar displacement predictions, with more variance in the range of values in the gradient boosted tree model. However, important distinctions between the modeling approaches show there is a much greater performance increase in the Wav2Vec-2.0 model. The self-supervised Wav2Vec-2.0 model was provided all data without excluding specific signals or tailoring the training data for the known nonstationarity by temporally sampling for a specific distribution. The data for the gradient boosted tree approach required special treatment to carefully account for nonstationarity in the time series and excluded 30 second time windows containing local earthquakes. The key important

difference is the Wav2Vec-2.0 model is trained using data from all stations together and is able to generalize for the solutions presented. In contrast, attempts to build a unified gradient boosted tree model using all data from all stations was largely unsuccessful. The gradient boosted tree model was trained for each station separately and was unable to converge or provide a general prediction when trained using all the available data[24]. Moreover, when considering the data availability and overall performance, the Wav2Vec-2.0 model results rival the best laboratory shear model predictions for both decision tree and deep learning models, where training data generally contains many more slip events and contains reduced variance due to the controlled laboratory conditions[34].

Both the gradient boosted tree and Wav2Vec-2.0 model show an upper prediction threshold for the final 10–20% of the loading cycles that does not match the GNSS (Fig. 4A). This output is not well understood and various attempts to further improve the predictions in the near failure time intervals were unsuccessful. At this time we speculate the plateau results from increased "noise" in the system since this is a volcano caldera collapse, not a traditional earthquake fault. The additional signals introduced to the model would contain tremors and potentially other fluid related processes that the model causes the model to saturate and is unable to fully capture the ground motions.

To further test how the self-supervised Wav2Vec-2.0 model generalizes input waveforms for multiple downstream tasks, we present a unique result showing the prediction for the onset of slip during each collapse event (Fig. 5 where "a" though "t" are labeled in Fig. 3A). The event slip timing predictions are quite good and shows a decrease in displacement at the slip event, excluding the final event with no change predicted. The blue line represents the prediction at each time step, i.e., each dot is the value representing 30 seconds obtained using the previous 300 seconds of data. The selection criteria is observational; we are looking for a decrease in displacement at the prescribed time step or a decrease that matches the GNSS measurements. The inclusion of events that match the GNSS (e.g., "g", "l", "q", and "r") arises from the potential timing discrepancy introduced by the 30 second windowing of the displacement data. The model detects the onset of slip and correctly predicts the decrease in displacement as observed in the GNSS measurements for 17 of the 20 test cases with 30 second temporal resolution. In two slip events ("c" and "e") the model incorrectly predicts the timing by one interval indicating the seismic waveforms contain the larger amplitude signals associated with slip. Only one slip event ("t") is completely missed by the model at the end of the sequence.

**Future predictions.** The Wav2Vec-2.0 model is reconfigured and trained to determine if the seismic waveform data contain a signature of future behavior, i.e., will the model predict if the failure event is going to occur in the next 30 second time window. This is in contrast to the contemporaneous predictions in Fig. 5. For the future predictions, seismic waveforms are input to the model using a window of time-history and the label is the future time segment of displacement[15]. For example, 0-300 seconds of continuous seismic data are used as the model input with the target label from the displacement at 300-330 seconds.

Just as the contemporaneous predictions, the input-output window pair selection were tested using a grid search (Supplementary Fig. S2). The best performing model uses a 300 second input window length for the subsequent 30 second window for the future prediction. The continuous time series results have a $R^2$ value of 0.62 and are similar to the contemporaneous predictions (Supplementary Fig. S3). Interestingly, the $R^2$ values for the input-output window pair selection (Supplementary Fig. S2) are on-average higher the contemporaneous predictions (Supplementary Fig. S1) for the training data set but quite similar for the testing data. The future predictions appear good throughout the earthquake cycle (Supplementary Fig. S3), however, the $R^2$ value may be misleading as shown in the expanded view for the earthquake failures. The future timing of the onset of slip is correctly predicted for only 3 out of 20 events (Fig. 6)–future slip "h", "i", and "p"– with the remaining showing the decrease in displacement in the following time interval. There appears to be prediction of the onset of (partial) slip in other events, e.g., "m" and "n", but that is a subjective to the criteria for selection.

The delayed timing of predictions indicate this model is far less reliable and only predicts the slip after the failure event is observed in the data. As a test if data leakage associated with the model prediction timing, we conducted model tests using future predictions for two successively later time intervals. Those results clearly indicate slip is only predicted after the model input contains the slip event of interest. For this data the model fails to extract information that describes the short term future behavior and directly contrasts the performance of the contemporaneous results. Additional work is required to further address this issue but is beyond the scope of the results presented.

## Discussion

Previous applications using data from a laboratory shear experiment show that decision tree and deep learning approaches are capable of mapping the contemporaneous fault friction directly from the

broadcast, continuous seismic signal[1,2]. In a major step forward in advancing the temporal and spatial scale from the laboratory to field applications, we recently showed that by applying decision tree models to the Kīlauea volcano data, the recorded seismic signals broadcast during faulting associated with the 2018 caldera collapse sequence can be mapped to the contemporaneous GNSS surface displacement[24]. In the present work, a pre-trained audio speech recognition model is fine-tuned for seismic applications. The results are improved over our previous work using decision tree model, successfully showing that seismic signals emitted during the 2018 Kīlauea caldera collapse sequence contain signatures of the fault displacement during the loading cycles (Fig. 3). Importantly, with the improved self-supervised modeling approach we successfully demonstrate the ability to predict the onset of a slip event (Fig. 5), which is not possible with a decision tree model.

The waveform data selection process was extensively tested to understand the temporal resolution. We conduct a grid search of input-output time-window pairs (Supplementary Fig. S1) and find the best input-output window pair using the $R^2$ values is 270-30 seconds. While the contemporaneous model predictions for 270 seconds input window progressively degrade with longer output window lengths, the predictions are acceptable out to the maximum output window attempted of 300 seconds. While longer input windows perform better in general, when using the shorter input-output pairs, e.g., a 30-30 seconds, the predictions are also acceptable. Indeed, one may expect longer input time series produce better predictions but this is not the case. Even short input windows acceptably predict contemporaneous displacement out to 300 seconds.

Perhaps the most surprising result is the prediction of all but 3 of the 20 earthquake failures in the test data (Fig. 5). To our knowledge, these predictions represent the most robust, contemporaneous slip-timing predictions of seismogenic faults. The results indicate that a modified version of Wav2Vec-2.0, originally developed to analyze acoustic signals, in combination with the transformer-based model we previously developed to analyze laboratory shear experiments[15], is highly effective in contemporaneous prediction of surface displacement during the Kīlauea sequence, but not the near future displacement. Unfortunately, little, if any, evidence was observed describing the displacement in future time intervals, indicating the waveforms may not contain this signature or the model is inadequate for this challenging task (Fig. 6). Our view is that the question of near-future prediction of earthquake timing remains unresolved and that new models with different data may resolve this question.

The analysis at Kīlauea is an important step forward in our understanding of the workings of fault slip across orders of magnitude in spatial and temporal scales. It is gratifying that the same statistical features found in the laboratory[1], Cascadia[16,35], and the San Andreas Fault[18] were also identified and applied by the decision tree model at Kīlauea[24]. These applications demonstrate the processes captured by the machine learning model span length scales of more than 10 orders of magnitude. We do not know what features were used by the modified Wav2Vec-2.0 model for the present work, but believe they may be similar based on all of the previous fault-characteristic prediction work applying decision trees at Kīlauea[24] and laboratory and slip predictions conducted by us and others. The work has important implications because it tells us the methods that have been developed for the laboratory and simulation shear experiments can work in active fault zones.

A fundamental goal is developing data-driven machine learning approaches for determining the slip-timing of faults with intervals of decades to hundreds of years. We note the prediction problem is also being addressed be applying machine learning to develop high fidelity earthquake catalogs[36]. We are in the process of addressing the problem by developing several promising approaches, including applying fault simulations at the scale of active fault zones[37] for model training as we

have successfully done in laboratory scale studies[19]. The remaining prediction metrics—magnitude and location—remain exceedingly challenging, although we, along with collaborators, are working on the magnitude prediction question, and some progress has been made previously by our group in laboratory studies[16] and work by colleagues[5]. The location question has not been addressed by data driven approaches as far as we are aware. One can imagine however, that back-projection methods applying continuous seismic data streams may advance the question of where an event may take place, along with existing geologic, geophysical and historical data sets.

## Methods

### Seismic and GNSS data

The Hawai'ian Volcano Observatory operates a permanent seismic network with multiple stations collocated with high-rate GNSS receivers (Fig. 1). The analysis is performed using data recorded between 1 June 2018 and 2 August 2018 when approximately 50 ~ $M_w$5 earthquakes occurred as the caldera step-wise collapsed. The seismic–GNSS stations pairs RIMD–CRIM, BYL–BYRL, and UWE–UWEV are on the caldera rim and AHUD–AHUP is south of the caldera with the AHUP GNSS monument located about 0.5 km away from the seismic sensor.

The high-rate GNSS solutions are processed relative to station KOSM with 5 second resolution[38]. The displacement time series for the east and north GNSS components are combined to a horizontal magnitude as $\sqrt{east^2 + north^2}$. The long-period ground motion is defined by applying a filter with a width of 10 days and this trend is removed to focus only on displacements associated with successive collapse events (1-3 day repeat interval). The GNSS data are downsampled by averaging 6 consecutive windows to obtain a 30 second displacement sampling-rate time-series. To reduce high frequency fluctuations, the period between each collapse is smoothed using a 5 minute window.

The continuous 3-component broadband (HH channel at 100 Hz) seismic records are high quality without large intervals of missing data. For each trace the instrument response is deconvolved to velocity, detrended, filtered between 0.1-50 Hz, and merged into a continuous time series. The number of missing data points is negligible and an interpolated value is inserted to maintain the sample rate. The data traces are sliced into 30 second (3000 points) non-overlapping windows that are temporally consistent with the GNSS time series interval so that each 30 second seismic waveform segment corresponds to the ground displacement measurement.

### Wav2Vec-2.0 model

**Self-supervised pre-training**. The modified Wav2Vec-2.0 deep learning model extends our previous effort applying laboratory data to predict near-future fault friction using a convolution encoder-decoder transformer model[15] to the more challenging task of surface displacement prediction on a seismogenic fault. For this task we introduce a self-supervised learning technique developed for automatic speech recognition into the updated workflow. Specifically, we apply the Wav2Vec-2.0 model[32] as described briefly in the introduction section, and in detail as follows.

The application here exploits the strength of Wav2Vec-2.0 that uses self-supervised learning of latent vector representations from raw audio waveforms. The model outperforms supervised and semi-supervised methods in the task of speech recognition applying limited amounts of labeled data. Our task is to determine if the model can successfully extract latent vector representations from raw waveforms of continuous seismic emission from the sequence of collapse events at Kīlauea, and to determine if these extracted latent vectors can predict ground displacement in the immediate future. To accomplish this downstream task, we sequentially attach our previously designed transformer decoder model[15] following the Wav2Vec-2.0 model and fine-tune the entire model with limited amounts of displacement data.

Here we use the HuggingFace's Transformers Library[39] and adopt all the model configurations and training hyperparameters in the HuggingFace's Transformers Library "speech-pretraining" example. The Wav2Vec-2.0 model consists of a feature encoder with 7 temporal convolutional layers. We apply the 12-transformer-layer BASE configuration described in ref. 32. The self-supervised learning task is, for each window of input waveform, approximately 49% of its data points are masked, and the model learns to reconstruct the masked portion from the visible portion. The waveform is encoded into a sequence of vectors in a simultaneously trained codebook. Instead of directly reconstructing the waveform data, the model tries to select the correct latent vectors for the masked portion from 100 candidate vectors. The total number of trainable parameters in the model is 95.05 million. The loss consists of a contrastive loss associated with the self-supervised learning task, and a diversity loss for the learnable codebook.

We combine all training waveforms (approximately 16 days of data) from the four seismic stations (RIMD, UWE, BYL, AHUD) as our pre-training data set, and combine all testing waveforms (approximately 16 days of data) from the four seismic stations for validation. Each sample in the batch is comprised of 300 seconds recorded at 100 Hz (30,000 data points) with 3 channels (east-north-vertical) for 90,000 data points. Assuming a batch size of 8, the shape of the input batch tensor is [8 (batch size), 30,000 (data points), 3 (channels)], we pass the waveform of each channel independently to the Wav2Vec 2.0 model by reshaping the input batch to [24 (waveform samples) = 8 (batch size) * 3 (channels), 30,000 (data points)]. Each waveform is standardized to zero mean and unit variance individually. This per-sample standardization performs better than per-dataset standardization. The learning rate is linearly increased to 32,000 training steps up to 0.001 and then linearly decays. The AdamW optimizer[40] is applied. The model is trained with 8 Nvidia Tesla V100 GPUs on one node. The batch size per GPU is 8, resulting in an effective batch size of 64 per step. Model training performance uses the masked reconstruction accuracy on validation waveform data after each training epoch. The model with the lowest validation contrastive loss is saved. Early stopping is applied when the validation loss does not reduce for 10 epochs.

**Fine-tuning pre-trained model and Wav2Vec-2.0 predictor**. The model output is produced using a transformer decoder we refer to as the Wav2Vec-2.0 predictor. We use a standard fine-tuning procedure for Wav2Vec2.0 that freezes the feature extractor portion containing 7 convolutional layers, and the transformer encoder portion of the model is fine-tuned together with our downstream transformer decoder. Specifically, the pre-training model coefficients are fine-tuned using the input waveforms and target displacements recorded at seismic station RIMD and GNSS station CRIM. We apply the same layer structure as used in Wav2Vec-2.0.

A transformer decoder model[30] is designed for contemporaneous and near-future displacement predictions using the Wav2Vec-2.0 as a signal encoder. An input window of time length $t_{in}^{hist}$ contains 3 channels of seismic waveform history, for example, using $t_{in}^{hist} = 300s$ and batch size of 8, the input tensor shape is [8 (batch size), 30,000 (data points), 3 (channels)]. The input waveforms are reshaped to [24 (waveform samples), 30,000 (data points)] as in the pre-training step and then processed by the pre-trained Wav2Vec-2.0 model to obtain the latent representation vectors for each individual waveform with a shape of [24 (waveform samples), 93 (time steps), 768 (vector dimension)]. These vectors are projected onto an vector quantized embedding space with a dimension size of 256 for a tensor shape of [24, 93, 256], which is the key/value vectors input to the transformer decoder. The transformer decoder consists of multi-head-attention and feed-forward network layers, with the modification of applying layer normalization first and using a Gaussian error linear unit (GELU) activation function. We tested the number of successive layers for 1, 2,

3, 4 and found the number does not impact the results. In the case of future prediction, the sampling rate of the target displacement is 0.0333 Hz (every 30 seconds), an output window of time length $t_{out}^{future}$ requires predicting $t_{out}^{future}/30s$ future data points. For example, when using $t_{out}^{future} = 60s$ to predict 2 future displacements at 30 seconds and 60 seconds respectively, we input a query vector of shape [24, 2, 256] to the transformer decoder. The query vector is from a learnable positional embedding layer with a maximum length of 10, to note the maximum output length of 300 seconds in this work. Therefore the output from the decoder is the same shape of [24, 2, 256]. It is reshaped to 3 channels for each input waveform [8 (batch size), 3 (channels), 2, 256] and then channel-averaged to the shape [8, 2, 256]. A 10% dropout layer and a linear layer project the last dimension of the vector to the final displacement output with shape [8, 2, 1]. The loss function applied is the mean squared error (MSE). The optimizer parameters and the computing settings are identical to the pre-training. The number of trainable parameters in the model is 91.42 million. We use the model with the best $R^2$ and early-stop the fine-tuning if the validation loss does not improve for 10 epochs.

### Grid search for the input/output time lengths

We systematically test the contemporaneous and near-future prediction capabilities of Wav2Vec-2.0 predictor by applying different input and output window lengths using station RIMD–CRIM data. The window lengths range between 30 to 300 seconds, with a step size of 30 seconds. Training Wav2Vec-2.0 predictors starts with the same pre-trained Wav2Vec-2.0 model and is fine-tuned using combinations of input/output window length pairs as denoted in the format Xs-Ys. For the grid search, we first fine-tune ten models to predict the future displacement at 30 seconds with input seismic windows Xs-30 seconds, X = 30, 60, 90, …, 270, 300; as shown in the first column of Supplementary Fig. S1. Wav2Vec 2.0 predictor provides the flexibility of the output window size by changing the query positional embedding vector into the transformer decoder, e.g., shape [24, Y/30, 256], while maintaining the same model architecture. Hence for a fixed input length X, we can train the subsequent nine increasing output lengths Y = 60, 90, …, 270, 300 in an iterative manner as shown for each row in Supplementary Fig. S1. In this procedure, we start with the fine-tuned Xs-30 seconds model weights and use the trained weights of the Xs-Ys model to initialize the training of the subsequent Xs-(Y+30)s model. This repeated for the range of time intervals tested.

## Data availability

All seismic waveform data is publicly available through the Earthscope consortium SAGE facilities for the HVO network[41]. All GNSS data is publicly available through the Earthscope consortium[38].

## Code availability

The software for generating the map in Fig. 1A is publicly available at www.generic-mapping-tools.org. The Wav2Vec-2.0 model is publicly available at https://huggingface.co/facebook/wav2vec2-base. All code employed for the analysis of data herein is available upon request to the corresponding authors.

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

## Acknowledgements
This material is based upon work supported by the U.S. Department of Energy, Office of Science, Office of Basic Energy Sciences, Geosciences program under Award Number LANLE3CB. Approved for unlimited release LA-UR-24-28409.

## Author contributions
C.W.J. Prepared the data, performed preliminary analysis, interpreted results, and wrote initial manuscript and revised. K.W. applied Wav2Vec-2.0 model, interpreted results, and revised manuscript. P.A.J. interpreted results and revised manuscript.

## Competing interests
The authors declare no competing interests.

## Additional information

**Peer review information** : *Nature Communications* thanks Sadegh Karimpouli, Leonard Seydoux and the other, anonymous, reviewer(s) for their contribution to the peer review of this work. A peer review file is available.

