## [Transparent Peer Review file · Nature Communications]

Automatic Speech Recognition Predicts Contemporaneous Earthquake Fault Displacement

Corresponding Author: Dr Christopher Johnson

Version 0:

Reviewer comments:

Reviewer #1

(Remarks to the Author)

Dear Editor,

The manuscript presents a study on fault displacement prediction using advanced speech detection neural networks, which is described as a pioneering effort in the application of machine learning to seismology. This work builds on a prior study by Johnson and Johnson (2024). While the results do not demonstrate a significant improvement in prediction accuracy, they highlight the potential of speech detection networks to analyze continuous seismic signals, opening up promising avenues for future research in this field. As such, I strongly support its publication. My minor comments are as follows:

- Figure 3: It is evident that in all displacement cycles, the predictions tend to be overestimated during the first half of the cycle and underestimated during the second half. This pattern is also observed in the results produced by GBT. How do the authors account for this behavior? Is not it the effect of the activation function in the last layer? If yes, other functions such as 'sigmoid' or 'tanh' may enhance it.

- Lines 372-377: The authors state that "the Wav2Vec 2.0 model results rival the best laboratory shear model predictions...". It may cause misunderstanding, as Rouet-Leduc et al. (2017) and several subsequent studies have improved laboratory results by up to 90%. I suggest removing this statement.

- Figure 5: A. If we consider the start of the 'vertical dark gray bar' as the onset of failure, panels 'f, g, l, m, q, and r' are not in agreement with others (this also applies to Figure 6). B. Following my earlier comment, the underestimation near failure leads to the prediction of relatively small displacements compared to the larger GNSS displacements. In a real-world application of this network, such low displacements might not be recognized as indicating the onset of failure, as seen in panels 'a, d, o, and s'.

- Figures S1-S3: How do the authors explain the higher R^2 values for future predictions (67-77%) compared to contemporaneous predictions (49-63%), which contradicts the description in the text (lines 593-603)? Furthermore, the highest R^2 in Figure S2 is 77%, while in Figure S3, it is 62%. Clarification on this is needed.

- Normalization: In line 780, it is mentioned how the waveform (input) is standardized. How about the outputs (displacements)? If min-max normalization is used, discuss about the problems in real applications, where the range of displacements may exceed the range of training data.

Best

Sadegh Karimpouli
GFZ Potsdam

Reviewer #2

(Remarks to the Author)

Dear Authors and Editor,

Please find below my review of the paper "Automatic Speech Recognition Predicts Contemporaneous Earthquake Fault Displacement," submitted to Nature Communications by Dr. Johnson and colleagues. Overall, I find the manuscript to be of high quality and believe it deserves publication. I have included only minor comments that I hope will help further strengthen the manuscript and foster continued discussion.

The authors present an application of machine learning for seismic-to-displacement signal prediction in the case of the 2018

Kīlauea crater collapse. Building on prior applications across diverse seismic cycles—experimental fault systems, slow-slip events in Cascadia—the authors extend their methods to this new context, demonstrating the robustness of their approach. Notably, they highlight that models trained in related domains can effectively capture seismic signal features necessary for predicting contemporaneous fault displacement, with the Wave2Vec 2.0 model proving particularly effective. An ambitious extension explored here is near-future displacement prediction, which, although not fully successful in this study, holds strong potential for future refinement. This suggests that continued advances in AI-based speech processing could significantly impact the field by enabling models with enhanced predictive capabilities. The introduction effectively presents the challenges posed by limited observational cycles in field data and the motivation for this work.

My first point of discussion is the issue of non-stationarity in seismic cycles. In Figure 1, for instance, the absolute northward displacement at the CRIM station appears to saturate slightly, while eastward displacement does not. More importantly, this effect only affects the testing dataset. The long-period displacement amplitudes exceed the dynamic stick-slip motion, prompting the authors to remove this long-term trend to achieve stationarity. This decision raises several follow-up questions: (1) Could the displacement rate serve as an alternative dataset that balances stationarity with prediction accuracy? While calculating the derivative would increase noise, it might also retain critical non-stationary components to be predicted. (2) Given that training on a small, evolving segment of data may limit predictive accuracy, could this trend discrepancy between training and testing data affect generalization? (3) Are comparable non-stationary signals observed in lab simulations? If so, this might clarify whether these non-stationary elements hinder model predictions in real-world applications.

Figure 3 reveals a notable pattern: predicted displacement saturates, forming a plateau as the model approaches the time-to-failure (visible in Figures 3A and C). This persistent issue, observed across several of the authors' prior studies, suggests a gap in model performance that resists improvement even with advanced features delivered by Wave2Vec. Do you have insights on this plateau in the context of Kīlauea? I understand that answering this question may require an in-depth feature analysis, which the authors note as an objective for future research.

Another interesting aspect of the study is the use of multiple seismic stations, which incorporates both spatial and temporal features. Although the methods section is comprehensive, it remains unclear whether the authors aggregated the traces in time (e.g., creating combined 4x300-second channel traces) or "vertically" (i.e., 4x3 channels of 300 seconds each). I am unsure if such a distinction would affect the model's performance, but this clarification could help readers better understand the model's approach to data structuring. A follow-up question arises: could model performance improve with advanced waveform combination techniques, such as cross-correlation functions or beamforming?

Finally, I wonder if narrowing the frequency band of analysis could improve the model's performance. Currently, the study uses the broadband signal (from 0.1 to 50 Hz), which includes a substantial amount of information that may not be related to the crater collapse (e.g., oceanic microseismic noise).

I remain available for further discussions and would be glad to read a revised version of the manuscript if needed.

Sincerely,
Leonard Seydoux

Reviewer #3

(Remarks to the Author)

The manuscript describes an approach in which a machine learning tool developed by FAIR (Facebook AI Research) called Wav2Vec 2.0 for speech recognition is here repurposed to predict earthquake fault displacement from continuous seismic data. The study uses data from the 2018 caldera collapses at the Kīlauea volcano in Hawai'i, based on about 40 collapses within about 30 days, intervalled by slow reload. Continuous seismic record was done from a network of broadband stations which recorded moderate magnitude seismic events up to about Mw 5. The authors train Wav2Vec 2.0 to predict "Contemporaneous Displacement" as well as "Future Displacement" from seismic waveforms, fitting it with observed GNSS data. The method is a small improvement compared to recently published work on GRL, 2024, where the same data were predicted using a gradient boosted tree approach. Here they achieve an R^2 of 0.63 in predictions, while in the study published on GRL they had obtained $R^2 = 0.59$.

The authors highlight that their results show matching patterns between the observed real-world seismic record and the laboratory recording of controlled experiments, which suggest that the successes of machine learning in predicting laboratory events should be mirrored in real world correct predictions. However unfortunately, while they show a robust ability to detect "contemporaneous displacement" (I don't like their use of the word predict, which I would only reserve for "future" predictions) in seismic data, they are very honest at highlighting how Wav2Vec performs poorly when trying to predict future events.

Overall, I find that although their results are interesting and their study stimulating future research, I do not believe that their progress is substantial compared to previous works. Here below I list more in detail the issues that I find in the manuscript:

Comments:

1) The data challenge is not described in quantitative detail. They only emphasize the challenges of applying machine learning to geophysical data with limited historical and comprehensive event coverage. Better knowledge of the computational training and testing time of the use of Wav2Vec vs other algorithms (e.g. gradient boosted tree) might

reinforce the argument whether these should be or not adopted in the future, for example for instantaneous detection of the caldera movements, as the authors show their approach is successful in that.

2) The discussion on data leakage could be expanded for clarity. The authors highlight that future displacement predictions are less reliable due to data leakage and model limitations. But in fact, what seems to me, "data leakage" is a term used to cover the fact that the algorithm "predicts" the slip after it has "detected" it. Maybe the authors intended something more sophisticated than that, and in this case how it could be mitigated, but for what results from this work, there is no real "prediction", even short term one, happening from this approach.

3) Lack of clarity on how the 3D data of a broadband station is turned into a 1D Wav2Vec latent space vector as described. The authors describe in the technical part that they combine 8 sensors times 3 channels, i.e. a 24 dimensional vector times 30k data points (100Hz for 300s), which are initially expanded to 93x768 (71424) sized latent vector and then reduced to 93x256 (23808). I find the 93 timesteps hard to comprehend. What is the origin of this number? Symmetrically, I do not understand how optimal is 768. I understand that these numbers might not be relevant overall and that results would have likely be similar with other choices of the shape of the latent spaces, but still the rationale behind this specific choice should be given.

4) The work compares (paragraph starting in 348) the results of a recently published GRL paper (Seismic Features Predict Ground Motions During Repeating Caldera Collapse Sequence) where the same data were used to train and predict displacement using the gradient boosted tree approach. However, they do not show clearly what the improvement was. From the text one reads that R2 increased from 0.59 to 0.63 by changing the method, but that does not seem such a drastic improvement. Overall, the authors explain at length the superiority of Wav2Vec in terms of being able to combine in a unique prediction all the 3x8 channels of data, however from this point of view, I would say that the improvement has been insufficient and one is confronted with the question whether actually these 24 channels offer more information than a single one. More in general, the authors do not offer an answer to the question that given the greater dataset used, why isn't the performance of Wav2Vec much better than gradient boosted tree?

5) The text uses both the terms "future displacement" and "failure prediction," without an explicit definition. While as a reader I have an intuitive understanding of the fact that the first is the quantitative observed displacement in the following 30 seconds, while the second attempts at identifying the caldera collapses, I believe that the authors should have given clear definitions.

Version 1:

Reviewer comments:

Reviewer #1

(Remarks to the Author)

Dear Editor,

Following the responses provided in the response file and the revisions made to the manuscript, I support the publication of the manuscript.

Best regards,
Sadegh Karimpouli

Reviewer #2

(Remarks to the Author)

Dear Authors and Editor,

I have carefully reviewed the insightful responses provided by the authors regarding the remarks I made on the manuscript. I appreciate the thorough answers and the related corrections, and I agree with all points addressed. Therefore, I fully support the publication of the paper in its current form.

Best regards,
Leonard Seydoux

Reviewer #3

(Remarks to the Author)

I believed that the authors have addressed all the points that I made and I recommend their work for publication

REVIEWER COMMENTS

Reviewer #1 (Remarks to the Author):

Dear Editor,

The manuscript presents a study on fault displacement prediction using advanced speech detection neural networks, which is described as a pioneering effort in the application of machine learning to seismology. This work builds on a prior study by Johnson and Johnson (2024). While the results do not demonstrate a significant improvement in prediction accuracy, they highlight the potential of speech detection networks to analyze continuous seismic signals, opening up promising avenues for future research in this field. As such, I strongly support its publication.

We thank you for the time spent reviewing our manuscript and the constructive feedback. During the revisions we have taken the opportunity to further explain why these results are a significant advancement, primarily considering this is a self supervised model and insensitive to the nonstationarity that required additional processing in the previous work. We now state, with additional added text “The limited ability of gradient boosted tree models to properly generalize due to the non-stationarity is problematic if trying to extract salient features directly from the data for downstream tasks as is done with a foundation model.”

Provided are response to each comment and our best effort to address the point or concern cited.

My minor comments are as follows:

- Figure 3: It is evident that in all displacement cycles, the predictions tend to be overestimated during the first half of the cycle and underestimated during the second half. This pattern is also observed in the results produced by GBT. How do the authors account for this behavior? Is not it the effect of the activation function in the last layer? If yes, other functions such as ‘sigmoid’ or ‘tanh’ may enhance it.

This observation was a topic of much discussion while preparing the manuscript. This is not a function of an activation function in the final layer. In Johnson and Johnson (2024) this same feature is observed in the displacement predictions. In that study a boosted tree model was used, and models were developed using a data set that was reduced by excluding time windows containing an earthquake phase arrival (EQtransformer detection algorithm was applied to label time windows with P/S arrival). Since the seismicity rates increase throughout the loading cycle there was less temporal sampling at the time nearest failure. However, the results were largely unchanged and the displacement prediction still contain a similar plateau (see Fig. 4). We provide the reader with more context in the results we now state “Both the gradient boosted tree and Wav2Vec 2.0 model show an upper prediction threshold for the final 10-20% of the loading cycles that does not match the GNSS (Fig. \ref{F4}A). This output is not well understood and various attempts to further improve the predictions in the near failure time intervals were unsuccessful. At this time we speculate the plateau results from increased “noise” in the system since this is a volcano caldera collapse, not a more traditional earthquake fault. The additional signals introduced to the model would contain tremors and potentially other fluid related processed creating specific conditions in the loading rate that the model is unable to fully capture.”

- Lines 372-377: The authors state that "the Wav2Vec 2.0 model results rival the best laboratory shear model predictions...". It may cause misunderstanding, as Rouet-Leduc et al. (2017) and several subsequent studies have improved laboratory results by up to 90%. I suggest removing this statement.

We have revised the statement to clearly convey the performance improvements when comparing to previous laboratory based analysis. We now state “Moreover, when considering the data availability and overall performance, the Wav2Vec 2.0 model results rival the best laboratory shear model predictions for both decision tree and deep learning models, where the data generally contains many more slip events and less variance due to the controlled laboratory conditions”

- Figure 5: A. If we consider the start of the ‘vertical dark gray bar’ as the onset of failure, panels ‘f, g, l, m, q, and r’ are not in agreement with others (this also applies to Figure 6). B. Following my earlier comment, the underestimation near failure leads to the prediction of relatively small displacements compared to the larger GNSS displacements. In a real-world application of this network, such low displacements might not be recognized as indicating the onset of failure, as seen in panels ‘a, d, o, and s’.

The vertical line is the selection window we prescribe using the timing estimate we obtain from the data. Using an observational perspective, we are not assigning a pass/fail criteria based on the amplitude of change. We note that in this real world application having a small displacement might not warrant a failure alert, however we are testing if the information is obtainable from the data in the model framework presented. For a more concise describing of the properly detected events we now state “The selection criteria is observational; we are looking for a decrease in displacement at the prescribed time step or a decrease that matches the GNSS measurements. The inclusion of events that match the GNSS (e.g., ‘g’, ‘l’, ‘q’, and ‘r’) arises from the potential timing discrepancy introduced by the 30 second windowing of the displacement data.”

- Figures S1-S3: How do the authors explain the higher R^2 values for future predictions (67-77%) compared to contemporaneous predictions (49-63%), which contradicts the description in the text (lines 593-603)? Furthermore, the highest R^2 in Figure S2 is 77%, while in Figure S3, it is 62%. Clarification on this is needed.

This is a good observation that we did not fully explore because the main test with the prediction of a large slip event. We note that this is specific to the training data because the testing data has very similar R^2 values. We now state “Interestingly, the R^2 values for the input-output window pair selection (Fig. S2) are on-average higher the contemporaneous predictions (Fig. S1) with the training data set but quite similar with the testing data. The displacement predictions appear good throughout the earthquake cycle (Fig. S3), however, the R^2 value may be misleading as shown in expanded view for the earthquake failures.”

- Normalization: In line 780, it is mentioned how the waveform (input) is standardized. How about the outputs (displacements)? If min-max normalization is used, discuss about the problems in real applications, where the range of displacements may exceed the range of training data. The model output is standardized for stability but reported with the variance and mean corrected back to the original scale. In this application the prediction is a continuous variable that can scale beyond the data, however we did not further explore those limitations.

Best

Sadegh Karimpouli

GFZ Potsdam

Sadegh, we again thank you for your time reviewing the manuscript.

Reviewer #2 (Remarks to the Author):

Dear Authors and Editor,

Please find below my review of the paper “Automatic Speech Recognition Predicts Contemporaneous Earthquake Fault Displacement,” submitted to Nature Communications by Dr. Johnson and colleagues. Overall, I find the manuscript to be of high quality and believe it deserves publication. I have included only minor comments that I hope will help further strengthen the manuscript and foster continued discussion.

The authors present an application of machine learning for seismic-to-displacement signal prediction in the case of the 2018 Kīlauea crater collapse. Building on prior applications across diverse seismic cycles—experimental fault systems, slow-slip events in Cascadia—the authors extend their methods to this new context, demonstrating the robustness of their approach. Notably, they highlight that models trained in related domains can effectively capture seismic signal features necessary for predicting contemporaneous fault displacement, with the Wave2Vec 2.0 model proving particularly effective. An ambitious extension explored here is near-future displacement prediction, which, although not fully successful in this study, holds strong potential for future refinement. This suggests that continued advances in AI-based speech processing could significantly impact the field by enabling models with enhanced predictive capabilities. The introduction effectively presents the challenges posed by limited observational cycles in field data and the motivation for this work.

We thank you for the time spent reviewing our manuscript. We found the constructive feedback very helpful and have provided a response to each comment with our best effort to address the point or concern cited.

My first point of discussion is the issue of non-stationarity in seismic cycles. In Figure 1, for instance, the absolute northward displacement at the CRIM station appears to saturate slightly, while eastward displacement does not. More importantly, this effect only affects the testing dataset. The long-period displacement amplitudes exceed the dynamic stick-slip motion, prompting the authors to remove this long-term trend to achieve stationarity. This decision raises several follow-up questions: (1) Could the displacement rate serve as an alternative dataset that balances stationarity with prediction accuracy? While calculating the derivative would increase noise, it might also retain critical non-stationary components to be predicted. (2) Given that training on a small, evolving segment of data may limit predictive accuracy, could this trend discrepancy between training and testing data affect generalization? (3) Are comparable non-stationary signals observed in lab simulations? If so, this might clarify whether these non-stationary elements hinder model predictions in real-world applications.

The decision to remove the nonstationary component arose from our focus on the individual repeating slip events within the longer term deformation signal. This is an excellent question because, as noted by the reviewer, we are not prescribing the model to learn the full dynamics of the system. It is possible that the displacement rate could be incorporated into the dataset as an additional target label, however, that was not within the scope of the goals in this study. In our previous work we needed to perform a different scheme for the train/validate/test dataset due directly to this nonstationarity. Here, the Wav2Vec model can perform remarkably better without special consideration for this known quantity in the data. When comparing to laboratory data

sets, there are some experiments with long term trends in the data, but typically the need for additional preprocessing is not required.

Figure 3 reveals a notable pattern: predicted displacement saturates, forming a plateau as the model approaches the time-to-failure (visible in Figures 3A and C). This persistent issue, observed across several of the authors' prior studies, suggests a gap in model performance that resists improvement even with advanced features delivered by Wave2Vec. Do you have insights on this plateau in the context of Kīlauea? I understand that answering this question may require an in-depth feature analysis, which the authors note as an objective for future research.

Reviewer #1 also noticed the same pattern, and we repeat our reply here. This observation was a topic of much discussion while preparing the manuscript. This is not a function of an activation function in the final layer. In Johnson and Johnson (2024) this same feature is observed in the displacement predictions. In that study a boosted tree model was used, and models were developed using a data set that was reduced by excluding time windows containing an earthquake phase arrival (EQtransformer detection algorithm was applied to label time windows with P/S arrival). Since the seismicity rates increase throughout the loading cycle there was less temporal sampling at the time nearest failure. However, the results were largely unchanged and the displacement prediction still contain a similar plateau (see Fig. 4). We provide the reader with more context in the results we now state “Both the gradient boosted tree and Wav2Vec 2.0 model show an upper prediction threshold for the final 10-20% of the loading cycles that does not match the GNSS (Fig. \ref{F4}A). This output is not well understood and various attempts to further improve the predictions in the near failure time intervals were unsuccessful. At this time we speculate the plateau results from increased “noise” in the system since this is a volcano caldera collapse, not a more traditional earthquake fault. The additional signals introduced to the model would contain tremors and potentially other fluid related processes creating specific conditions in the loading rate that the model is unable to fully capture.”

Another interesting aspect of the study is the use of multiple seismic stations, which incorporates both spatial and temporal features. Although the methods section is comprehensive, it remains unclear whether the authors aggregated the traces in time (e.g., creating combined 4x300-second channel traces) or “vertically” (i.e., 4x3 channels of 300 seconds each). I am unsure if such a distinction would affect the model's performance, but this clarification could help readers better understand the model's approach to data structuring. A follow-up question arises: could model performance improve with advanced waveform combination techniques, such as cross-correlation functions or beamforming?

The model input is the 3-component 30 second waveform segments from all stations for model training. The data is structured such that a batch contains N , where the batch size is 8, waveform examples of 3-comp waveforms are for a tensor of shape $[N, 3, 30,000]$. For pretraining the input is reshaped to $[24, 30,000]$ so each waveform channel is input independently in the self-supervised pretraining. We now state more clearly “Assuming a batch size of 8, the shape of the input batch tensor is $[8$ (batch size), $30,000$ (data points), 3 (channels)]. We pass the waveform of each channel independently to the Wav2Vec 2.0 model by reshaping the input batch to $[24$ (waveform samples) = 8 (batch size) * 3 (channels), $30,000$ (data points)].”

Beamforming has not been applied here, but we did use that approach in a previous study (see Umlauf et al., 2024 on glacier motions) using the beamformed catalog as a unique model input.

That work focused on glacial sliding processes and the beamforming results were found to be a high ranking feature in the importance. Inputting station cross correlations is the topic of ongoing work in group now and results are showing success. Both great questions.

Finally, I wonder if narrowing the frequency band of analysis could improve the model's performance. Currently, the study uses the broadband signal (from 0.1 to 50 Hz), which includes a substantial amount of information that may not be related to the crater collapse (e.g., oceanic microseismic noise).

This is a good observation. Our thoughts is the time period considered (about 2 months) should not be largely affected by the known lower frequency signals which change periodically throughout the year and during large weather events. Our reasoning for including these lower frequency bands was to not constrain the model to a priori assumptions about important frequency content. Our justification is the model applied learned CNN filters that will account for a large range of frequencies when inputting the data to the transformer block. We now state "The hierarchical feature extraction captures information at different levels of abstraction, enabling the model to learn both low-level and high-level representations across a broad frequency spectrum of the input signal."

I remain available for further discussions and would be glad to read a revised version of the manuscript if needed.

Sincerely,
Leonard Seydoux

Leonard, we again thank you for your time reviewing the manuscript.

Reviewer #3 (Remarks to the Author):

The manuscript describes an approach in which a machine learning tool developed by FAIR (Facebook AI Research) called Wav2Vec 2.0 for speech recognition is here repurposed to predict earthquake fault displacement from continuous seismic data. The study uses data from the 2018 caldera collapses at the Kīlauea volcano in Hawai'i, based on about 40 collapses within about 30 days, intervalled by slow reload. Continuous seismic record was done from a network of broadband stations which recorded moderate magnitude seismic events up to about Mw 5. The authors train Wav2Vec 2.0 to predict “Contemporaneous Displacement” as well as “Future Displacement” from seismic waveforms, fitting it with observed GNSS data. The method is a small improvement compared to recently published work on GRL, 2024, where the same data were predicted using a gradient boosted tree approach. Here they achieve an R^2 of 0.63 in predictions, while in the study published on GRL they had obtained $R^2 = 0.59$.

The authors highlight that their results show matching patterns between the observed real-world seismic record and the laboratory recording of controlled experiments, which suggest that the successes of machine learning in predicting laboratory events should be mirrored in real world correct predictions. However unfortunately, while they show a robust ability to detect “contemporaneous displacement” (I don't like their use of the word predict, which I would only reserve for “future” predictions) in seismic data, they are very honest at highlighting how Wav2Vec performs poorly when trying to predict future events.

Overall, I find that although their results are interesting and their study stimulating future research, I do not believe that their progress is substantial compared to previous works. Here below I list more in detail the issues that I find in the manuscript:

We thank you for the time spent reviewing our manuscript and the constructive feedback. In particular we have addressed the concern raised about the progress compared to previous works. We now include more detailed description of using self-supervised learning models instead of the previous efforts applying supervised learning techniques. In the last paragraph of the introduction we now state “The widespread adoption of unsupervised, or self-supervised, learning techniques for signal classification has not seen the same rise in popularity for seismic waveform analysis as compared to other deep learning fields; despite intriguing results demonstrating the ability of ML models to separate signals with no a priori knowledge [e.g., 26–28]. The lack of generalization in supervised learning models does not facilitate fine-tuning an existing model for out-of-distribution data, and instead requires large amounts of labeled training data to develop a new model for a specific application [29]. The supervised learning is now eclipsed by the development of self-supervised foundation models using unlabeled data to produce generative pre-trained transformer networks [30].”

We then further describe the limitation in the previous work and include more details throughout for this advancement.

Below we provide a response to each comment.

Comments:

1) The data challenge is not described in quantitative detail. They only emphasize the challenges of applying machine learning to geophysical data with limited historical and comprehensive

event coverage. Better knowledge of the computational training and testing time of the use of Wav2Vec vs other algorithms (e.g. gradient boosted tree) might reinforce the argument whether these should be or not adopted in the future, for example for instantaneous detection of the caldera movements, as the authors show their approach is successful in that.

The lack of data for a full earthquake cycle is comprehensively described in Wang, Johnson, Bennet, and Johnson (2024) Nature Communications. In that we present a transfer learning approach that could be implemented for decadal to century time scales that are common for recurrence times of earthquake faults. We now more clearly state “The primary challenge for applying machine learning to frictional failure on most faults hosting large magnitude earthquakes is the long repeat times, ranging from decades to thousands of years, and therefore, the lack of geophysical instrumental data that capture a complete loading cycle, often only a small fraction of a loading cycle.”

As for the computation expense and training time, yes, these modeling efforts require HPC GPU clusters to facilitate model development, training, validation, and test. Specifically, this compute cost scales with the size of the data. In this work the expense compounds when testing the optimal input and output time windows which are an important result. This is a necessary resource and a requirement to further test what types of model architectures are most applicable.

We fully acknowledge these resources are not available at every research institute, but external options exist, e.g., AWS. This is not meant as a criticism but instead we use the analogy of a geochemist in a research institute with a \$1M mass spectrometer to do her radiological dating studies. It's the correct resource for the work performed. As for adopting these types of models for future use, that is exactly what we aim to present. The goal of our research group is to determine the performance metrics with the ambition of developing a foundation model as has been successful in computer vision and natural language processing. We now state to clearly articulate this point “The limited ability of the gradient boosted tree model to properly generalize due to the non-stationarity is problematic if trying to extract salient features directly from the data for downstream tasks as is done with a foundation model. In the present work we develop a self-supervised learning approach to determine if contemporaneous and future predictions can be made directly from the continuous seismic data.”

2) The discussion on data leakage could be expanded for clarity. The authors highlight that future displacement predictions are less reliable due to data leakage and model limitations. But in fact, what seems to me, “data leakage” is a term used to cover the fact that the algorithm “predicts” the slip after it has “detected” it. Maybe the authors intended something more sophisticated than that, and in this case how it could be mitigated, but for what results from this work, there is no real “prediction”, even short term one, happening from this approach.

Point well taken and we have expanded our comments on the ability to make short term predictions. We now state “The delayed timing of predictions indicate this model is far less reliable and only predicts the slip after the failure event is observed in the data. As a test if data leakage associated with the model predicting timing, we conducted model tests using future predictions for two successively later time intervals. Those results clearly indicate slip is only predicted after the model input contains the slip event of interest. For this data the model fails to extract information that describes the short term future behavior and directly contrasts the performance of the contemporaneous results. Additional work is required to further address this

issue but is beyond the scope of the results presented.”

3) Lack of clarity on how the 3D data of a broadband station is turned into a 1D Wav2Vec latent space vector as described. The authors describe in the technical part that they combine 8 sensors times 3 channels, i.e. a 24 dimensional vector times 30k data points (100Hz for 300s), which are initially expanded to 93x768 (71424) sized latent vector and then reduced to 93x256 (23808). I find the 93 timesteps hard to comprehend. What is the origin of this number? Symmetrically, I do not understand how optimal is 768. I understand that these numbers might not be relevant overall and that results would have likely be similar with other choices of the shape of the latent spaces, but still the rationale behind this specific choice should be given.

The model input is the 3-component 30 second waveform segments from all stations for model training. The data is structured such that a batch contains N, where the batch size is 8, waveform examples of 3-comp waveforms are for a tensor of shape [N, 3, 30,000]. For pretraining the input is reshaped to [24, 30,000] so each waveform channel is input independently in the self-supervised pretraining. We now state more clearly “Assuming a batch size of 8, the shape of the input batch tensor is [8 (batch size), 30,000 (data points), 3 (channels)]. We pass the waveform of each channel independently to the Wav2Vec 2.0 model by reshaping the input batch to [24 (waveform samples) = 8 (batch size) * 3 (channels), 30,000 (data points)].”

The other dimensions arise from the wav2vec2 architecture as described in the original paper. The original model has the feature encoder that contains seven temporal convolutions blocks. The paper states "The feature encoder contains seven blocks and the temporal convolutions in each block have 512 channels with strides (5,2,2,2,2,2,2) and kernel widths (10,3,3,3,3,2,2)." 93 is the number that the 30k timesteps waveform is reduced to 93 timesteps after this series of CNN layers. This is the shape of the intermediate tensor. The 768 is from the BASE version of the transformer model in the original paper which states "BASE contains 12 transformer blocks, model dimension 768, inner dimension (FFN) 3,072 and 8 attention heads." These are hyperparameters that can be tuned for the AE waves, but beyond the scope of this project, hence we just adopted the original model hyperparameters.

4) The work compares (paragraph starting in 348) the results of a recently published GRL paper (Seismic Features Predict Ground Motions During Repeating Caldera Collapse Sequence) where the same data were used to train and predict displacement using the gradient boosted tree approach. However, they do not show clearly what the improvement was. From the text one reads that R2 increased from 0.59 to 0.63 by changing the method, but that does not seem such a drastic improvement. Overall, the authors explain at length the superiority of Wav2Vec in terms of being able to combine in a unique prediction all the 3x8 channels of data, however from this point of view, I would say that the improvement has been insufficient and one is confronted with the question whether actually these 24 channels offer more information than a single one. More in general, the authors do not offer an answer to the question that given the greater dataset used, why isn't the performance of Wav2Vec much better than gradient boosted tree?

We have revised the Section “Contemporaneous Displacement Prediction” to more clearly state the improvements, notably the ability of the model to generalize. The added text now includes these points:

(1) “Quantitatively the increase in R^2 could be considered quite modest, qualitatively we show a large improvement in the range of values at the lower displacement prediction which

occur following a slip event (Fig. \ref{F4}B). One would not expect the R^2 value to solely capture the performance change considering the overall variance in the training data (Fig. \ref{F1}) and the inherent point-to-point fluctuations in GNSS high rate measurements.”

(2) “The self-supervised Wav2Vec 2.0 model was provided all data without excluding specific signals or tailoring the training data for the known non-stationarity by temporally sampling for a specific distribution. The data for the gradient boosted tree approach required special treatment to carefully account for non-stationarity in the timeseries and excluded 30 second time windows containing local earthquakes.”

(3) “In contrast, attempts to build a unified gradient boosted tree model using all data from all stations was largely unsuccessful. The gradient boosted tree model was trained for each station separately and was unable to converge or provide a general prediction when trained using all the available data”

(4) “Provided the self-supervised Wav2Vec 2.0 model generalizes the input waveforms for multiple downstream tasks we shown a unique result for the prediction for the onset of slip during each collapse event”

5) The text uses both the terms "future displacement" and "failure prediction," without an explicit definition. While as a reader I have an intuitive understanding of the fact that the first is the quantitative observed displacement in the following 30 seconds, while the second attempts at identifying the caldera collapses, I believe that the authors should have given clear definitions. Throughout the revision process we have made an effort to streamline and define the terms used, specifically with respect to failure, slip, predictions, etc.